# *Brachypodium* Antifreeze Protein Gene Products Inhibit Ice Recrystallisation, Attenuate Ice Nucleation, and Reduce Immune Response

**DOI:** 10.3390/plants11111475

**Published:** 2022-05-31

**Authors:** Collin L. Juurakko, George C. diCenzo, Virginia K. Walker

**Affiliations:** 1Department of Biology, Queen’s University, Kingston, ON K7L 3N6, Canada; george.dicenzo@queensu.ca (G.C.d.); walkervk@queensu.ca (V.K.W.); 2Department of Biomedical and Molecular Sciences, School of Environmental Studies, Queen’s University, Kingston, ON K7L 3N6, Canada

**Keywords:** *Brachypodium distachyon*, cold acclimation, antifreeze proteins, ice-binding proteins, leucine-rich repeats, protein modelling, protein–protein interactions, protein-peptide interactions, ice nucleation, immune response

## Abstract

Antifreeze proteins (AFPs) from the model crop, *Brachypodium distachyon*, allow freeze survival and attenuate pathogen-mediated ice nucleation. Intriguingly, *Brachypodium* AFP genes encode two proteins, an autonomous AFP and a leucine-rich repeat (LRR). We present structural models which indicate that ice-binding motifs on the ~13 kDa AFPs can “spoil” nucleating arrays on the ~120 kDa bacterial ice nucleating proteins used to form ice at high sub-zero temperatures. These models are consistent with the experimentally demonstrated decreases in ice nucleating activity by lysates from wildtype compared to transgenic *Brachypodium* lines. Additionally, the expression of *Brachypodium* LRRs in transgenic *Arabidopsis* inhibited an immune response to pathogen flagella peptides (flg22). Structural models suggested that this was due to the affinity of the LRR domains to flg22. Overall, it is remarkable that the *Brachypodium* genes play multiple distinctive roles in connecting freeze survival and anti-pathogenic systems via their encoded proteins’ ability to adsorb to ice as well as to attenuate bacterial ice nucleation and the host immune response.

## 1. Introduction

Since they cannot escape, plants must cope with a multitude of stresses, no more so than when winter approaches and they must defend themselves against ice formation, dehydration, and the pathogens that thrive even under snow cover. Do these multi-pronged assaults require the synthesis of specialised single-function defensive proteins or has evolution selected for multi-functional protectors or moonlighting proteins [1]? More than two decades ago, Marilyn Griffith argued the latter, in that certain hydrolytic proteins could have dual functions; they were antipathogenic and could also protect against ice-mediated damage [2,3]. The proteins she investigated, however, did not have impressive antifreeze protein (AFP) activities and thus her observations were not fully embraced by the ice-binding protein field. However, AFPs from a grass, the false brome model cereal *Brachypodium distachyon* (hereinafter, *Brachypodium*), derive from a putative bi-functional post-translational product, making them an obvious target for the investigation of abiotic stress resistance and antipathogenic multifunctionality.

*Brachypodium* AFPs (*Bd*AFPs) are encoded by a family of ice-recrystallisation inhibition (IRI) genes, *BdIRI*s, with the translation products exported to the apoplast and processed by an apoplastic endopeptidase into two proteins: a leucine-rich repeat (LRR), derived from the amino-terminal domain, and a carboxy-terminal AFP [4,5,6]. Knockdown of the *BdIRI* translation products in transgenic *Brachypodium* as well as the heterologous expression of grass AFP in *Arabidopsis* have demonstrated that AFP activity confers protection against membrane electrolyte leakage subsequent to sub-zero temperature exposure [6,7]. Unlike AFPs from freeze-susceptible organisms, *Bd*AFPs do not lower the freezing point of solutions by more than ~0.1 °C, but they are very effective at IRI, or the prevention of ice crystal coalescence at high sub-zero temperatures or under freeze–thaw conditions, a property crucially important for freeze-tolerant grasses [6,8]. *Bd*AFP has been modelled based on its 67% identity to the AFP of the grass *Lolium perenne* (*Lp*AFP), and it is predicted to have a beta-solenoid structure with two flat ice-binding motifs [6,9]. Indeed, although it is an order of magnitude smaller (~13 kDa), *Bd*AFP is structurally similar to the concatenated beta-solenoid repeats modelled for the ~120 kDa ice-nucleation proteins (INPs) produced by ice nucleation-active plant pathogens, such as *Pseudomonas syringae* [6,10,11]. These bacteria can kill plants by inducing ice formation at high sub-zero temperatures in order to access intracellular nutrient stores, and they are of commercial concern since they are responsible for up to $7 billion in annual crop losses [12,13,14,15]. Notably, in addition to their IRI abilities, *Bd*AFPs appear to disrupt INP activity [8], but structural uncertainty of the large INPs have hitherto thwarted speculation about the interactions of these two proteins. 

Less is known about the LRR proteins derived from the *BdIRI*s. They have been modelled to the extracellular domain of the FLAGELLIN SENSITIVE2 (FLS2) receptor kinase of *Arabidopsis* [4,6]. FLS2 recognises the flg22 pathogen-associated molecular pattern (PAMP) derived from the flagellin of *P. syringae* and other bacteria, following which FLS2 triggers an immune response by initiating downstream biochemical changes and epigenetic reprogramming while energy is diverted from growth towards immune responses [16,17,18]. The similarity between the *BdIRI* LRR proteins and FLS2 suggested that the former can also bind flg22 and modulate the immune response. If so, the *BdIRI*s’ LRR domains could provide host protection against a costly defence response while the AFP domains neutralise one of *P. syringae’*s main weapons, INP. As noted, the structure and mechanisms of the molecular LRR:flg22 and AFP:INP interactions remain largely unknown. The recent release of AlphaFold now provides a tool to investigate these relationships in concert with new experimental evidence on the anti-pathogenic properties of the two translation products.

## 2. Results

### 2.1. AFP-Mediated Attenuation of INP-Induced Freezing and INP Models

The freezing point of *P. syringae* INP preparations was depressed by 1.55 °C with the addition of lysates from cold-acclimated (CA) wildtype *Brachypodium* compared to no lysate controls (*p* < 0.005, one way ANOVA; Figure 1; Appendix A). No changes in ice nucleation temperature were seen with non-acclimated (NA) lysates, which have no measurable AFP activity. Likewise, the nucleation temperatures of lysates from miRNA *BdIRI* knockdown lines (prOmiRBdIRI-1e or -3c [8]) with little to no AFP activity were not depressed, irrespective of their CA or NA treatment history, demonstrating that the attenuation of INP activity was dependent on AFP activity, as also shown by IRI assays (Figure 1d).

Attenuation of INP activity suggests a physical interaction between the two proteins, INP and AFP. Accordingly, in silico modelling and docking predictions were performed using AlphaFold [19] and FRODOCK, respectively [20,21]. Of the seven isoforms, the product of *BdIRI1* was selected as the primary modelling representative, but all other full sequence models were made (Appendix A), as were just the LRR domains (Appendix A) or just the AFP domains (Appendix A). Previously, Phyre2 homology models of *Bd*AFPs alone were folded according to the crystal structure of *Lp*AFP [7,11]. The AlphaFold-generated model of the *BdIRI* primary translation product shows that the LRR and AFP domains are connected via a disordered linker that would facilitate endoprotease cleavage following secretion to the apoplast (Appendix A). Similar to the Phyre2 model, this new model folds each of the AFP domains into right-handed beta-solenoids consisting of opposing repetitive ice-binding a- and b-faces, with the sequence N**xVx**G/N**xVx**xG, where x is an outward-facing, hydrophilic residue and the boldface font indicates conserved triplets implicated in ice-association (Figure 2a,b and Appendix A).

Similarly, *P. syringae* INP (InaZ variant) was modelled as a representative INP. The bulk of the model shows a twisted ~28.5 × 0.25 nm beta-solenoid with dual, opposing flat water-organising surfaces characterised by ~63 tandem GYGS**TQT**AxxxS**xLx**A repeats, where x is an outward-facing hydrophilic residue, the presumptive water-ordering conserved triplets are shown in bold, and the putative inter-strand dimerisation tyrosine-ladder triplets are underlined (Figure 3c and Appendix A). The opposing water-ordering surfaces appear to make equivalent contributions to INP activity [22,23] and both appeared as flat a- and b- faces in the model. As shown, an N-terminal membrane anchor connects to the regularly ordered repeats of the beta-solenoid via a disordered linker composed of 61 residues (Figure 2c). The model also shows a cap-like structure at the carboxyl-end of the solenoid that terminates with a disordered tail of charged residues. The cap likely provides structural stability, preventing unwinding and amyloid fibril aggregation [24,25,26].

Both AFP and INP models show that the conserved triplets of the water/ice-associating surfaces have two ranks of outward-facing residues containing methyl groups, which are thought to be important for the arrangement of clathrate or ice-like water molecules (Appendix A) [22]. All of the top 10 scoring docking predictions show interactions through the a-faces of *Bd*AFP and INP (Figure 2d,e) with docking scores as high as 7212 (Appendix A). In contrast, the closely related *Lp*AFP, previously shown to attenuate INP less effectively than *Bd*AFP [9], showed a lower maximum docking score of 4203, with 20% of the models showing INP interactions on the single ice-binding a-face [27], and 80% on the opposite face (Appendix A). A fish Type III AFP with little ability to perturb INP activity [28,29] showed multiple possible interactions and a still lower maximum score of 2783 (Appendix A).

Previous homology modelling of INPs based on the large bacterial repeats-in-toxin (RTX) proteins followed by manual inspection suggested that head-to-tail INP dimers could form through tyrosine ladders [10]. Here, FRODOCK and AlphaFold modelling applied to short INPs composed of eight tandem repeats, necessitated by computational restrictions, also showed similar dimerisation (Figure 3a). To experimentally test this INP model, ice nucleation assays were performed. Heating INPs at 37 °C for 24 h depressed the freezing point by 3.33 °C, suggesting that the proposed inter-strand interactions may not fully reform (Appendix A), which is consistent with the requirement for low temperature activation [30]. The polyphenol, tannic acid (TA), destabilises tyrosine ladders such as those found in amyloid fibres [31], and thus INA assays were performed with TA as a further test of the modelled inter-strand binding. As hypothesised, TA addition significantly depressed ice nucleation 2.28 °C more than INP preparations alone (*p* < 0.001, one way ANOVA; Figure 3b). Significantly, when *Bd*AFPs and TA were added together, the attenuation of INP activity was greater than with TA alone (2.88 °C depression compared to control INP experiments; *p* < 0.001, one way ANOVA), suggesting that the “spoiling” of ice nucleation by TA and AFPs was likely via separate sites. Again, this supports our *Bd*AFP:INP interactive models. Based on models for the INP monomer (Figure 2c) and the modelled INP:INP interactions (Figure 3a), a representation of the INP oligomer filaments was made, mediated through the formation of tyrosine ladders between adjacent monomers (Figure 3d,e). These oligomers were then used to form aggregate sheets stabilised by interactions between positively and negatively charged residues (Figure 3f and Appendix A), consistent with the nucleation theory-predicted aggregation of 34 INPs required to mediate freezing at high sub-zero temperatures [32].

### 2.2. LRR-Mediated Attenuation of the Host Immune Response and Modelling

Representative LRR sequences corresponding to *BdIRI1*, *3*, and *7* were expressed in *Arabidopsis*, and the impact on the native immune response was assessed using oxidative burst assays (Figure 4a). *Arabidopsis* lines expressing any of the LRRs showed significantly impaired responses (*p* < 0.001, one way ANOVA) when leaf disks were exposed to the immunogenic flg22-γ peptide [33], compared to wildtype *Arabidopsis* controls or transgenic plants bearing empty plasmids. Mean photon counts were reduced by 71% and 76% with LRR1, by 79% and 83% with LRR3 and by 64% and 70% with LRR7, when compared to transgenic and wildtype controls, respectively. No response to the flg22-α peptide was seen in any of the samples, as was expected given that this peptide is non-immunogenic to *Arabidopsis* [33]. A schematic (Figure 4b,c) illustrates our concept of LRR function in attenuating the host immune response.

The AlphaFold LRR1 model showed similarity to the FLS2 crystal structure [34] with opposing beta sheets and irregular alpha-helices forming a concave solenoid-like structure (Figure 5a). To further investigate the potential LRR:flg22 binding suggested by the experimental assays, AlphaFold interactions were modelled as described [35], which predicted binding of the flg22-γ epitope along the length of presumptive LRR binding pocket (Figure 5c). Five permutations of the flg22-γ peptide sequence were created, and modelling of these failed to show the same binding orientations or the hydrogen bonds identified in the authentic LRR:flg22 complex, adding further support for the interaction between flg22 and *Brachypodium* LRRs (Appendix A). PDBePISA was used to further interrogate the interface between LRRs, including the representative LRR1 and the flg22-γ or flg22-α peptides, which predicted that binding occurs through a hydrophobic interface as indicated by a negative solvation free energy gain of −3.6 and −2.1 kcal mol^−^^1^, respectively (Appendix A). The calculated interaction-specific surface areas for both peptides were substantive (*p* < 0.5; https://www.ebi.ac.uk/pdbe/pisa/ (accessed on 28 January 2022)), while interactions with permutations of the flg22-γ sequence ranged from 10–50% of the absolute value of the interface area between the LRR1:flg22-γ complex (Appendix A).

Using similar methodology, two LRRs together were also shown to have high interaction surface areas (Appendix A; Figure 6). For example, when LRR3 and LRR4 were allowed to bind in silico, the heterodimer showed a negative solvation free energy gain of −7.3 kcal mol^−^^1^ when then bound to the flg22-γ peptide, at least twice that of the LRR:flg22-γ interaction, suggesting that more than one LRR could simultaneously bind to flg22-γ *in planta*. Likewise, LRR3 and LRR4 homodimers showed negative solvation free energy gains of −9.1 kcal mol^−^^1^ and −11.9 kcal mol^−^^1^, respectively, with the LRR4 homodimer forming a sandwich around flg22-γ stabilised by 25 predicted hydrogen bonds. By taking this concept further, modelled tetramers showed similar results (Figure 6g,h). Such interactions were not seen with flg22-γ permutation controls (not shown).

The *Bd*PEPR1 (BRADI_5g15070v3) LRR ectodomain and *Bd*RLP23 (BRADI_5g13280v3) are distinct LRR proteins known to bind other ellicitors and not flg22. These LRR proteins were therefore modelled with flg22-γ in order to assess the specificity of the predicted interaction between the BdIRI-derived LRRs and flg22 (Appendix A; Appendix A). Significantly, these other LRRs showed unfavourable binding when modeled with either the flg22-γ and flg22-α peptide. *Bd*RLP23 showed a negative solvation free energy gain of −1.1 kcal mol^−^^1^ with flg22-γ and −1.0 kcal mol^−^^1^ with flg22-α, while *Bd*PEPR1 showed a negative solvation free energy gain of −1.5 kcal mol^−^^1^ with flg22-γ and 1.1 kcal mol^−^^1^ with flg22-α. Both *Bd*RLP23 and *Bd*PEPR1 showed below average hydrophobicity for the structures with flg22 peptides, suggesting that the surfaces were not specific and would be unable to attenuate a flg22-induced immune response. Taken together, these other LRRs further underscore *Bd*LRR:flg22-γ specificity.

## 3. Discussion

The coming of winter is accompanied by low temperature-affiliated stresses in temperate plants [36] including the orchestration of ice formation on *P. syringae’s* anchored INPs, presumably to freeze the apoplast at high sub-zero temperatures and thus allow nutrient access for the pathogen. Such destructive activity must be neutralised by the plant, with ice formation allowed at lower temperatures, and with additional safeguards in place to prevent ice crystal coalescence into larger membrane-damaging crystals. Additionally, light and water limitations would argue that energetically expensive immune responses must be muted [37,38]. We now show that all of these requirements can be met by the *BdIRI* gene family. Thus, Griffith’s hypothesis that plants encode proteins that are both antipathogenic and ice protecting is correct [2], but even she did not envision the remarkable attributes of the proteins encoded by *BdIRI*.

Investigations into how *Brachypodium* achieves these defences demand an insight into the physical interactions between proteins. However, INPs are a challenge due to their membrane association and their large ~120 kDa size. The previous RXT-homology folded INP model [10] is largely concordant with our new AlphaFold model, with a twisted ~28.5 × 0.25 nm right-handed beta-solenoid repeat bearing flat putative water-organising a- and b- faces on opposite sides and with an amino-terminal cap and carboxyl terminal anchor domain (Figure 3). Similar to the homology-folded model, the new model combined with the FRODOCK docking algorithm applied to short INPs of eight tandem repeats shows the potential for inter-strand bonds through tyrosine ladder formation, which was tested by heating as well as TA addition, presumably targeting the cross-strand ladder [31]. Both of these treatments resulted in diminished ice nucleating activity, consistent with the known importance of dimerisation for INP function, since by themselves, monomers do not nucleate ice until −25 °C [39]. The tyrosine ladder shows an additional hydrophilic residue connection after energy minimisation, suggesting that the tyrosyl hydroxyl group could function in conjunction with the flat faces (a-side; Appendix A), thus seamlessly extending the ice-organisation template. INP dimers must oligomerise to form INP filaments, and subsequently form aggregate sheets of aligned filaments that could be stabilised by electrostatic interactions of charged residues, which have been previously shown to be necessary for INP activity [40] and which were present in the model. Such aggregates would minimally consist of the 34 INP monomers with a surface area of 240 nm^2^, covering ~10% of the length of the bacterium, sufficient to reach a critical embryonic ice nucleus mass at −2 °C (Figure 3) [32].

How do *Bd*AFPs disrupt these vast INP water-organising templates, as demonstrated by the experimental attenuation of INP activity (Figure 1)? The docking models suggest that ranks of serine/threonine residues on the a-face of *Bd*AFP dock near the ranks of the water-organising threonines on INP a-faces and possibly hydrogen bond, although this cannot be confidently predicted with the docking algorithm (Figure 3 and Appendix A). The model further suggests that *Bd*AFP:INP interactions would not interfere with tyrosine ladder formation and the subsequent staircase architecture facilitating ice formation [28,36]. Nonetheless, since the docked *Bd*AFP appears close to the INP water-organising triplets, it is not surprising that the experimental freezing temperature was lowered. A distinct site is also supported by INP assays showing that the attenuation of INP activity was greater in the presence of both TA and *Bd*AFP than either alone. Substantive maximum docking scores bolster the model, with scores for *Bd*AFP:INP 2.6-fold higher than for INP:fish Type III AFP, which has more than 10-fold the thermal hysteresis antifreeze activity of *Bd*AFP [41]. Notably, tested fish AFPs show little or no inhibition of INP activity, supporting the observation that attenuation of INPs is not an intrinsic property of AFPs, but rather an AFP-specific trait [7,29]. The grass *Lp*AFP with somewhat reduced ability to perturb INP activity showed two docking orientations involving the single *Lp*AFP ice-binding face as well as the opposite flat face, in addition to docking scores between those shown by *Bd*AFP and fish AFP (Appendix A). When *Bd*AFP docks with the INP solenoids, we posit that ice formation initiates at slightly lower temperatures, reducing the probability that membranes surrounding the apoplast are breached, but then continues to function by protecting the plant against dangerous ice recrystallisation with its two-sided ice-binding faces, even as temperatures hover near 0 °C.

Although *Bd*AFP attenuates INP activity and thus eliminates a risk imposed by *P. syringae*, the presence of bacterial flagella alerts the plant to a menace that typically results in the activation of the FLS2 receptor kinase through the LRR extracellular domain, leading to the mitogen-activated protein kinase cascade and signalling to direct energy to defensive proteins and away from growth and maintenance [18,42]. The *BdIRI* genes likely evolved from an ancestral rice phytosulfokine receptor tyrosine kinase with homology to the FLS2 immune receptor [4]. *P. syringae-*type flagellin proteins, flg22-γ, are suggested to interact with *Brachypodium* LRRs based on both experimental studies and the docking models, likely as a means to attenuate immune responses (Figure 4 and Figure 5). In the future, additional experiments could validate direct LRR:flg22-γ protein–peptide interactions. Notably, however, amino acids involved in flg22:LRR interactions are not the residues reported for flg22:FLS2 [34], suggesting an independent mechanism. We speculate that the apoplast LRRs function as pattern receptor-like proteins, except that they lack the intracellular domain. With their affinity for flagellin, these LRRs are hypothesised to bind to flagella from *P. syringae* and other pathogens and therefore interfere with subsequent binding to FLS2. This LRR occupation of the flagella would then reduce flagella availability for binding FLS2 and attenuate downstream effects involving BAK1, BIK1, and the phosphorylation of respiratory burst oxidase homolog protein D (RBOHD) that normally leads to extracellular release of ROS. ROS not only targets invading pathogens, but may also damage host tissues (Figure 4b,c). Therefore, LRR would mute the *Brachypodium* immune response and its attendant energetic requirements and thus join other negative regulators of ROS production [43]. Indeed, compared to control plants, *BdIRI* knockdowns showed more cellular death post-infection and an overall higher susceptibility to pathogenic *P. syringae* [8].

It is possible that the LRRs work synergistically to interact with flg22. LRR3 and LRR4, both expressed in leaves [6] and with modelled disordered extensions from the top of the “uncapped” solenoid, were predicted to self-assemble with seven hydrogen bonds in a manner that aligned to models of FLS2 (Figure 6). Indeed, analogous capless LRRs have previously been seen to self-assemble [44]. The binding of flg22-γ to the LRR dimer was “tighter” than to single LRRs and in a similar orientation with hydrogen bonding as shown in flg22-FLS2 binding (Appendix A; Figure 6). As monomers or dimers or even tetramers, the LRR may attenuate an overactive antipathogenic response in *Brachypodium* after low temperature exposure. Additionally, our simulations using other LRRs involved in immunity that bind separate PAMP elicitors showed unfavourable binding, providing support for the specificity of the *BdIRI* LRR: flg22-γ binding, similar to FLS2. The requirement for at least partial immune suppression is likely temporary. Once freezing takes place, the absence of free water may make pathogens less threatening and certainly, by the end of the CA period, the above-ground microbial community in *Brachypodium* changes so that whereas *P. syringae* accounted for 8% of the taxa prior to transfer to 4 °C, 6 days later, this taxon had disappeared [45].

Taken together, the *BdIRI* gene products have demonstrated multiple functions that include anti-ice activity that partially “spoils” the ability of INP to form ice at very high sub-zero temperatures, strong IRI to protect plant membranes against freeze damage, and the attenuation of the plant immune response. With the demonstrated ability to curtail unnecessary activation of host plant defences, these sequences collectively present prime foci for the development of low temperature resilience in transgenic horticultural and agricultural crops. However, they also force us to re-evaluate an updated Beadle’s concept of one-gene, one-protein, one-function; here, the protein products encoded by *BdIRI* have three distinct functions, an amazing “jab-cross-lead hook” combination to defy pathogens and freezing conditions alike.

## 4. Materials and Methods

### 4.1. Protein Modelling and Interaction Predictions

INPs and AFPs were modelled using a version of AlphaFold (version 2.1.0) running on a Colab notebook using a high-RAM runtime provided by Colab Pro+ [19]. Protein–protein docking predictions between various AFPs and INPs were performed using FRODOCK [18,19]. LRRs and flg22 binding predictions were modelled using AlphaFold with a 30-glycine linker, connecting the amino terminal of the LRR to the flg22 peptide sequence as described [35], with the LRR carboxyl apoplast localisation signal sequences omitted to allow for more representative binding events. All signal peptides were predicted using SignalP (version 6.0) [46]. The LRR-linker-flg22 models were opened in PyMOL (version 2.4.1) and each was independently selected as a separate object and the linker was hidden. The LRR was then re-created as a closed surface to allow for binding pocket visibility. Any flg22 peptide secondary structures were hidden using the command “cartoon loop” and side chains were shown as sticks. Homo- and heteromeric LRR complexes were modelled using AlphaFold-Multimer [47,48] to model potential self-assembly of LRRs.

Interactions between flg22 peptides and additional LRR proteins that bind ellicitors other than flg22 were modelled using the same approach as described above for modelling the interactions between flg22 peptides and LRR monomers. These included *Bd*PEPR1 (BRADI_5g15070v3) and *Bd*RLP23 (BRADI_5g13280v3), which are the *Brachypodium* homologs of the *Arabidopsis* accessions, AT1G73080 and AT2G32680, respectively. To isolate the ectopic LRR of the PEPR1 accession, the sequence and an AlphaFold structure were separately aligned with the ectopic LRR crystal structure and sequence from *Arabidopsis* PEPR1 (PDB ID: 5GR8) [49] prior to modelling flg22 interactions.

Random permutations of flg22 were created using the Shuffle Protein tool [50]. Using the Protein Data Bank in Europe (PDBe) Protein, Interfaces, Structures, and Assemblies (PISA) tool (version 1.52), interfaces between LRRs and peptides were analysed and scored to quantify predicted binding affinity [51,52,53]. Chains were modified accordingly using the alter command and hydrogen bonds were predicted using the find polar contacts between chains command. The PyMOL commands used can be found in Appendix A. All sequences used for models can be found in Appendix A.

### 4.2. Construction of LRR Plasmids for Arabidopsis

Constructs of *BdIRI*s, synthesised by GeneART (Thermo Fisher Scientific, Waltham, MA, USA), were confirmed by Sanger sequencing (Platform de Sequencage, Laval, QC, Canada). Sequences containing LRR motifs confirmed by InterProScan [54] were selected and used to form constructs in the pCambia1305.1 plant expression vector (Marker Gene Technologies Inc., Eugene, OR, USA). Alignment and preparation of consensus reads from Sanger sequencing and alignment to reference sequences as well as in silico assembly of constructs was performed using Benchling (https://benchling.com/ (accessed on 19 November 2020)). Primers were designed using constructs containing 20 bp overlaps for Gibson assembly (Appendix A). Amplicons of *Bd*LRRs containing overlaps were gel extracted using GeneJET Gel Extraction Kits (Thermo Fisher Scientific, Waltham, MA, USA) and pCambia1305.1 vectors were prepared and subsequently digested using *Nco*I restriction endonuclease prior to purification using Monarch DNA and PCR Cleanup Kits (New England Biolabs, Ipswich, MA, USA). Vector and insert concentrations were estimated using a Synergy H1 microplate reader (BioTek Instruments, Inc., Winooski, VT, USA) with a Take3 Micro-Volume Plate (BioTek Instruments, Inc., Winooski, VT, USA) and assembled using NEBuilder HiFi DNA Assembly Master Mix (New England Biolabs, Ipswich, MA, USA). Transformations were carried out using 2 µL of 4× diluted assembly mixes into *Escherichia coli* DH5α cells, screened using colony PCR, and confirmed with Sanger sequencing. Positive clones were transformed into *Agrobacterium* AGL1 (Invitrogen, Carlsbad, CA, USA), and re-screened for the insert using colony PCR.

### 4.3. Transient Expression in Arabidopsis

Transformation for transient gene expression in *Arabidopsis* was performed as described [55]. *Arabidopsis* seeds were germinated and sown to the commercial potting soil Sunshine^®^ Mix #1 (Sun Gro^®^ Horticulture, Agawam, MA, USA) and grown at standard conditions for three weeks. Simultaneously, *Agrobacterium* AGL1 cultures containing constructed plasmids were used to streak plates of YEB-induced agar containing appropriate antibiotics [55] and grown for two days before being scrapped and washed in 500 µL of wash buffer (10 mM MgCl2, 100 µM acetosyringone). Resuspended cells were gently vortexed and 100 µL was taken and diluted 10 times with infiltration buffer (25% MS, 1% sucrose, 100 µM acetosyringone, 0.01% Silwet L-77) with a final OD_600_ ≥ 12. Cells were then further diluted to an OD_600_ = 0.5. Finally, resuspended *Agrobacterium* cells were infiltrated into three-week-old *Arabidopsis* leaves on the abaxial side using a sterile 3 mL syringe with gentle pressure applied to the adaxial side. The zone of infiltration was marked gently using a black marker and labels were added to pots for transformant tracking. Following infiltration, plants were maintained under direct light for 1 h, using a consumer grade grow-op unit (SunBlaster Holdings ULC, Langley, BC, Canada), to allow for drying of the leaves. Plants were then transferred to the dark for 24 h before returning to standard growth conditions in a climate-controlled chamber (see Section 4.4. Plant Material and Growth Conditions below) for three days. Plants were removed from the chamber and leaf discs were taken using a sterilised 3.5 mm diameter biopsy tool (Robbins Instruments, Chatham, NJ, USA).

### 4.4. Plant Material and Growth Conditions

*Brachypodium* seeds of an inbred line, ecotype Bd21 (RIKEN, Wakō, Japan) and two transgenic AFP temporal knockdown lines [8] were sown in commercial potting soil Sunshine^®^ Mix #1 (Sun Gro^®^ Horticulture, Agawam, MA, USA) and maintained at 4 °C in darkness for four days to synchronise germination. Seeds were moved to a climate-controlled growth chamber (Conviron GEN2000, Queen’s University Phytotron, Kingston, ON, Canada) at standard *Brachypodium* growth conditions of 20 h days at ~150 μmol m^−2^s^−1^ at 24 °C followed by 4 h periods with no light at 18 °C. Plants were fertilised bi-weekly using 10-30-20 Plant-Prod MJ Bloom (Master Plant-Prod, Brampton, ON, Canada). Cold-acclimated plants were moved to a separate chamber (Econair GC-20, Queen’s University Phytotron, Kingston, ON, Canada) maintained at 4 °C and given a shortened day cycle of 6 h of light (~150 μmol m^−2^s^−1^) and 18 h dark for 48 h. Non-acclimated plants remained at standard conditions.

For *Arabidopsis* cultivation, wildtype seeds of ecotype Col-0 were soaked in water and placed at 4 °C in darkness for one week to synchronise germination before being sown in commercial potting soil Sunshine^®^ Mix #1 (Sun Gro^®^ Horticulture, Agawam, MA, USA). Plants were grown in a climate-controlled growth chamber (Conviron GEN2000, Queen’s University Phytotron, Kingston, ON, Canada) under standard growth conditions of 16 h days at 22 °C with light at ~150 μmol m^−2^s^−1^ followed by 8 h of darkness at 20 °C. Prior to assay, plants were transferred to 15 °C for a day in the event that this facilitated proper folding of the LRR domains.

### 4.5. Preparation of Extracts and Apoplast Samples for AFP Activity

For the analysis of AFP activity in wildtype and transgenic *Brachypodium*, extracts were prepared using a modified version of one that has been previously described [56]. After acclimating three-week-old plants, 50 mg of leaf tissue was flash frozen with liquid nitrogen, ground into a fine powder, suspended in 400 µL of NPE buffer (25 mM Tris, 10 mM NaCl, pH 7.5, EDTA-free protease inhibitor tablets), and subsequently shaken for 4 h at 4 °C in the dark on a GyroMini nutating mixer (Labnet International Inc., Edison, NJ, USA). Samples were centrifuged at 13,000 × *g* for 5 min, chilled at 4 °C for 5 min, and the centrifugation and incubation were repeated. The supernatant was transferred to 1.5 mL tubes and centrifuged again at 13,000 × *g* for 5 min before being returned to 4 °C. Protein concentration was quantified using a Synergy H1 microplate reader (BioTek Instruments, Inc., Winooski, VT, USA) with a Take3 Micro-Volume Plate (BioTek Instruments, Inc., Winooski, VT, USA) using A_280_ at a standard of 1 absorbance unit = 1 mg mL^−^^1^. Samples were normalised and diluted as described prior to assays, which included inspection of ice crystal morphology, electrolyte leakage, and measurements of thermal hysteresis, all as described elsewhere [5]. AFP assessment using IRI assays [56] was carried out using plants prepared as indicated above. Apoplast was prepared essentially as previously recommended [57]. After pipetting 10 µL of the apoplast samples 1 m onto a dry ice-chilled glass cover slip, they were annealed at −8 °C for 18 h. Images were captured through cross polarising film at 10× magnification, both before and at the end of the annealing period, with the experiment repeated independently at least three times for each sample. 

### 4.6. Ice Nucleation Assay

Assay of bacterial ice nucleation activity in the presence of AFPs and extracts was carried out as previously described [9,28,58]. Briefly, 2 µL of sample, containing either purified AFPs (1 mg mL^−^^1^), concentrated *Brachypodium* extracts (40 mg mL^−1^), or tannic acid (100 mM) combined with *P. syringae* INP (Ward’s Natural Science, Rochester, NY, USA) (at a final concentration of 50 µg mL^−1^) or INP alone, was pipetted onto a polarised film in 10 replicates. After placing the film in a chamber, the temperature was lowered from −1 °C to −12 °C, at a rate of 0.2 °C min^−1^. Images along with the temperature were recorded every 60 s. The temperature at which 90% of the samples (T_90_) froze was considered to be the nucleation point. The logarithmic cumulative number of ice nuclei per mL in each sample (K(T)) was calculated as previously described [9] using Vali’s equation [58]:(1)$$K\left(T\right)=-ln\left(N\left(T\right)/N_{0}\right)\times V-1$$
where N(T) represents the number of unfrozen samples remaining at time *T*, $N_{0}$ represents the total number of samples, V represents the volume of the sample, and the log(K(T)) was taken to represent the logarithm of cumulative number of ice nuclei per mL. All 10 assays of each sample were repeated in triplicate.

### 4.7. Immune Response Attenuation ROS Burst Assay

Plant immune response was assayed similar to that described previously [59]. Briefly, discs of transgenic *Arabidopsis* leaves were excised using a sterile 4 mm biopsy punch (Robbins Instruments, Chatham, NJ, USA) and individually placed abaxial side down in sterile water (100 µL) in each well of a white sterile 96-well microplate. Plates were incubated overnight at room temperature and then sealed with parafilm and covered with aluminium foil. After incubation, the water was carefully removed using a multichannel pipette, and replaced with 100 µL of solution containing 100 µM luminol, 10 µg mL^−^^1^ horseradish peroxidase (HRP), and 100 nM flagellin epitope ellicitor (or a control). Assays used a Synergy H1 Microplate Reader (BioTek Instruments, Inc., Winooski, VT, USA) using 1 s integration time, 60 min read time, with 2 min intervals measuring luminosity. Purified flg22 epitopes (EZBiolab, Parsippany, NJ, USA) included flg22-γ (representative of the flg22 epitope of y-proteobacteria) and flg22-α (representative of the flg22 epitope of α-proteobacteria), which do and do not elicit an immune response in *Arabidopsis*, respectively [32]. The assays were conducted on extracts of wildtype Col-0, Col-0:*Bd*LRR1, Col-0:*Bd*LRR3, Col-0:*Bd*LRR7, and a Col-0:pCambia1305.1 empty vector control with both epitope treatments and non-epitope blank controls. All assays were performed in triplicate.

### 4.8. Statistical Analysis

One-way ANOVAs with post hoc Tukey’s tests were performed in R (version 4.1.1) using the package multcomp for compact letter displays of groups.

## Figures and Tables

**Figure 1 plants-11-01475-f001:**
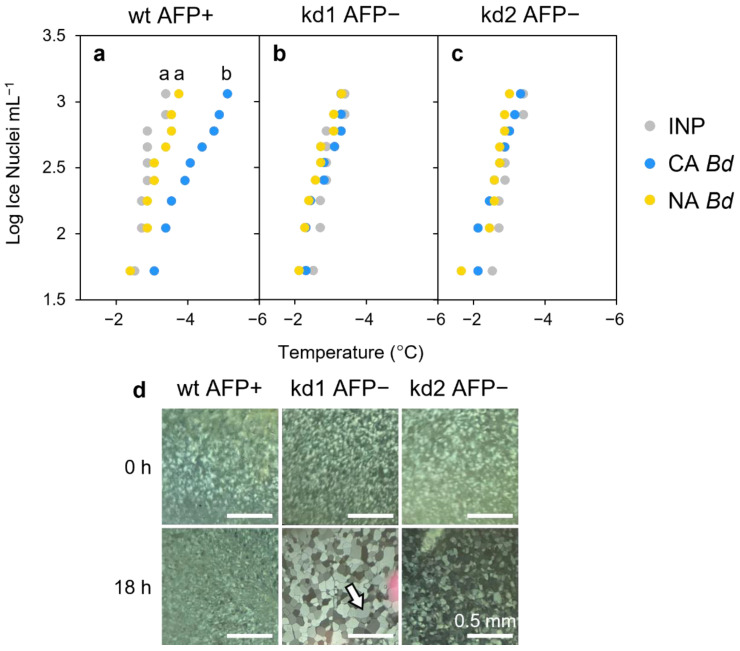
Impact of *Brachypodium* leaf tissue lysates or apoplast extracts on ice nucleation or ice crystal size. (**a**) Representative ice nucleation assays conducted using ice nucleating protein (INP) preparations (50 µg mL^−^^1^) and concentrated leaf lysates of *Brachypodium* leaf tissue (final concentration of 20 mg mL^−^^1^). Lowercase letter groupings indicating significance (*p* < 0.005, one way ANOVA) are shown as applicable. Samples (10 per plate) were repeated in triplicate with similar results. Comparison between INPs alone (grey dots) and INPs combined with lysates from cold-acclimated (CA; blue) or non-acclimated (NA; yellow) wildtype *Brachypodium* (Bd21). Notably, the CA leaf lysates were comparably effective as the −1.26 °C depression of INP activity by purified recombinant *Bd*AFPs at equal concentration [9]. (**b**) Comparison between INPs alone (grey dots) and INPs combined with lysates from the low temperature-induced transgenic *Bd*AFP knockdown line (prOmiRBdIRI-1e) that was either CA (blue) or NA (yellow). (**c**) Comparison between INPs alone (grey dots) and INPs combined with lysates from the low temperature-induced transgenic *Bd*AFP knockdown line (prOmiRBdIRI-3c) that was either CA (blue) or NA (yellow). (**d**) Ice-recrystallisation inhibition activity in diluted (0.01 mg mL^−1^) apoplast extracts from leaves of cold-acclimated wildtype plants (left) and reduced activity in two CA knockdown lines (prOmiRBdIRI-1e, middle, and prOmiRBdIRI-3c, right). Ice crystal sizes following an 18 h annealing period at −8 °C (lower images) are compared to those seen immediately after flash freezing (upper images). Scale bars represent 0.5 mm. The white arrow in the kd1 AFP− sample after annealing for 18 h indicates a single grown ice crystal for reference.

**Figure 2 plants-11-01475-f002:**
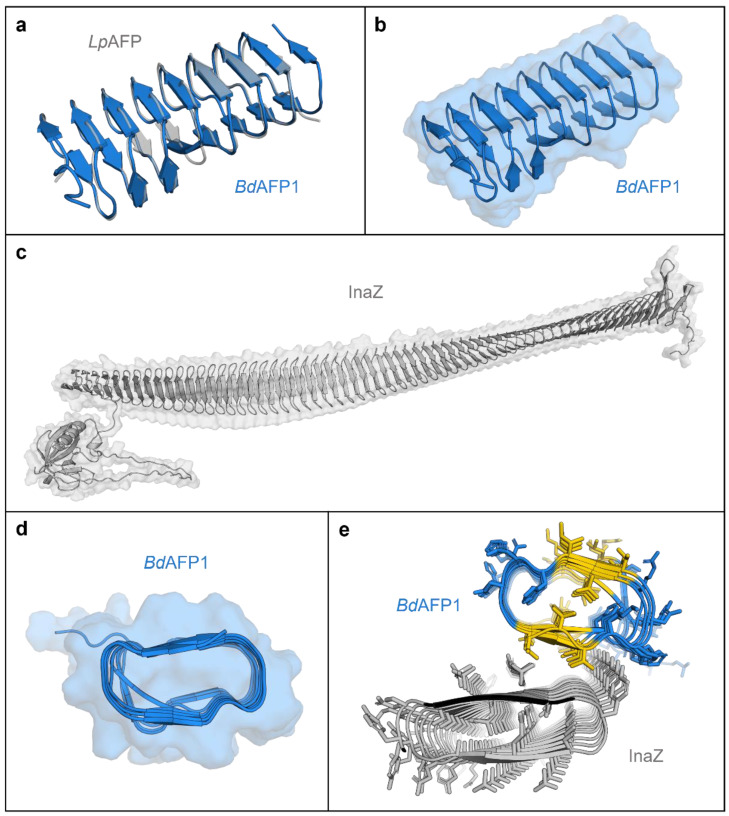
AlphaFold models of *Brachypodium distachyon* antifreeze protein (AFP) based on the representative *BdIRI1* AFP and *Pseudomonas syringae* ice nucleating protein (INP) and their interactions as predicted with FRODOCK. (**a**) AlphaFold *Bd*AFP model (in blue) shown aligned to the crystal structure of the related *Lp*AFP (PDB ID: 3ULT; in grey). (**b**) AlphaFold *Bd*AFP beta-solenoid fold with the solvent-accessible surface shown in transparency and with both ice-binding surfaces depicted as flat ribbons (with the “a- and b-faces” on the upper and lower surfaces, respectively, with residues detailed in Appendix A). (**c**) AlphaFold model of *P. syringae* INP as a twisted right-handed solenoid formed from the repetitive sequences with the N-terminal membrane anchor to the left, the cap sequence on the right, and the solvent-accessible surface area shown in transparency. (**d**) *Bd*AFP cross section demonstrating the hydrophobic internal core and showing the relatively flat surface contours created by the ice-binding motifs, with the “a-face” shown on the upper side. (**e**) Cross section of a FRODOCK docking prediction of interactions between *Bd*AFP (from *BdIRI1* AFP) in blue with the ice-binding faces in yellow (with the AFP “a-face”, closest to the INP), and *P. syringae* INP with two water-organising faces (with the INP “a-face” upwards, closest to the AFP). Note: The top 10 FRODOCK predictions (encompassing scores from 7212 down to 6349 as shown in Appendix A) show AFP binding along the entire length of the “a-face” of the INP via the “a-face” of the AFP, with only a representative image shown here.

**Figure 3 plants-11-01475-f003:**
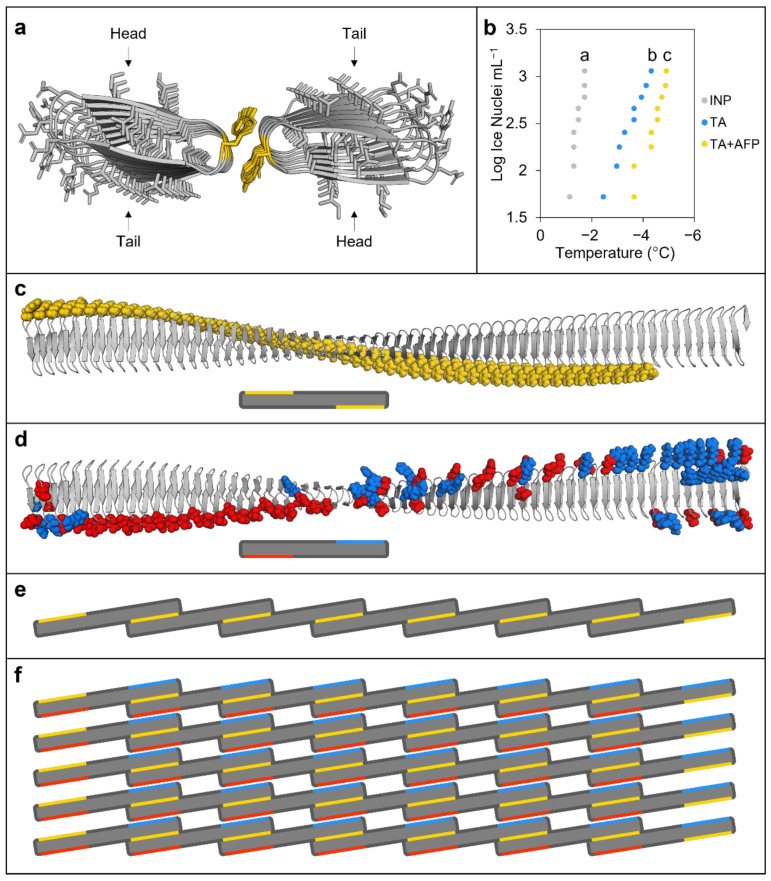
Modelling and other experimental evidence of ice nucleating protein (INP) interactions. (**a**) Tyrosine ladder interactions between INP monomers (InaZ) as predicted by FRODOCK using AlphaFold models, with tyrosine residues (in yellow) showing head-to-tail dimerisation. (**b**) Ice nucleation assay comparing INP preparations at 50 µg mL^−^^1^ alone (grey dots) or combined with either 100 mM tannic acid (TA; blue dots) or 100 mM TA and *Bd*AFP from cold-acclimated wildtype lysates (yellow dots). Significance (*p* < 0.01, one way ANOVA) compared to INPs alone is denoted by lowercase letters. Ice nucleation assays were repeated in triplicate with similar results. Note that the colligative depression of the TA solution was <0.2 °C. (**c**) An AlphaFold model of the *P. syringae* INP InaZ twisted beta-solenoid structure with the GPI-anchor and linker hidden, and the tyrosine ladder residues highlighted in yellow (top), shown together with a simplified illustration of the INP monomer with exposed tyrosine ladders represented by yellow bars (bottom). (**d**) InaZ with charged residues on the opposing side of the beta-solenoid to the tyrosine ladder, with negatively charged residues highlighted in red and positively charged residues in blue (top), shown together with a simplified illustration with charged residues represented by bars (bottom). (**e**) Schematic illustration of INP monomers (with the GPI anchor and cap sequence removed) organised by tyrosine ladder interactions, highlighted in yellow, to form a short INP filament. INP monomers would first partially dimerise with a single monomer forming tyrosine ladder interactions at each end of the twisted solenoid with two separate monomers on opposing sides. (**f**) Self-assembly of INP filaments would precede the assembly of parallel filaments into aggregate sheets, which could be stabilised through electrostatic interactions by the outward-facing positively and negatively charged residues (blue and red lines, respectively) found on opposing ends of the monomers on the side of the beta-solenoid opposite the tyrosine ladders (yellow lines). The schematic illustration shows 35 INP monomers (~250 nm^2^ of surface area) arranged into a sheet that would form patches on the surface of *P. syringae* (top down view). It should be noted that, in theory, 34 INP monomers must interact to reach a critical embryonic ice nucleus mass needed to achieve high sub-zero ice nucleation at −2 °C (see Section 3 Discussion).

**Figure 4 plants-11-01475-f004:**
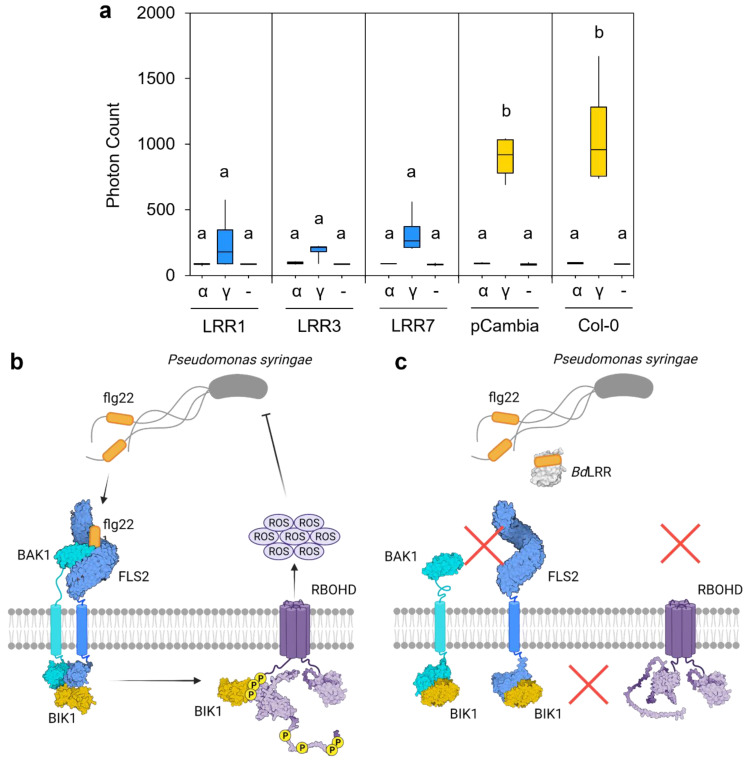
Interaction of *Brachpodium* leucine-rich repeat (LRR) protein sequences with bacterial flagellin protein (flg22). (**a**) Representative oxidative burst assays (reactive oxygen species) as assessed by the measurement of emitted photons as a proxy for *Arabidopsis* immune response. Results are shown for untransformed wildtype *Arabidopsis* (Col-0), Col-0 plants transformed with an empty vector (pCambia1305.1, shown as pCambia), and *Arabidopsis* lines expressing the *BdIRI1, BdIRI3*, and *BdIRI7-*encoded LRR protein products shown as LRR1, LRR3, and LRR7, respectively. The flg22 epitopes used were the flg22-γ (shown as γ) and flg22-α (shown as α) peptides previously shown to be immunogenic and non-immunogenic, respectively, to *Arabidopsis*. Blank controls consisting of no epitope (-) were also included. Lowercase letter groupings represent statistically significant differences (*p* < 0.001, one way ANOVA) and assays were repeated in triplicate with similar results. (**b**) Simplified illustration of how FLAGELLIN-SENSITIVE 2 (FLS2) induces the downstream release of extracellular ROS upon detection of flg22-γ epitopes. (**c**) Simplified illustration of how *Bd*LRRs may attenuate with the FLS2-dependent release of extracellular ROS in the presence of flg22-γ epitopes. Illustration made using BioRender.com.

**Figure 5 plants-11-01475-f005:**
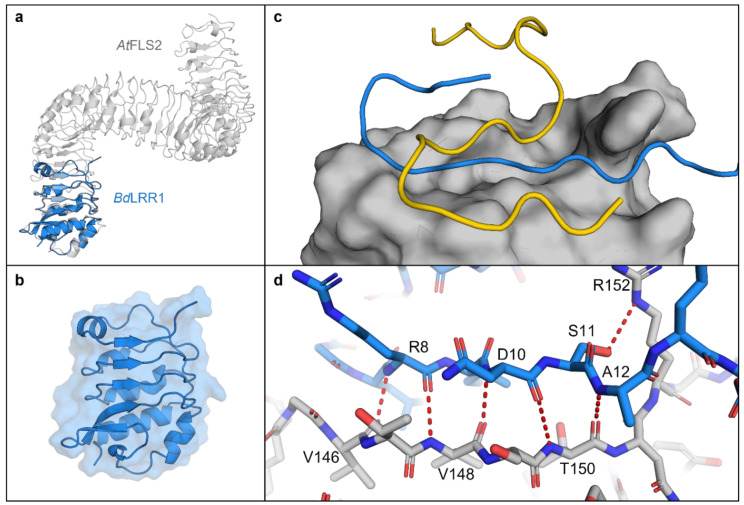
AlphaFold models of the *Brachypodium distachyon* LRR1 domain (based on *BdIRI1*). (**a**) LRR1 in blue, overlayed on the crystal structure of the *Arabidopsis thaliana* extracellular receptor FLS2 (PDB ID: 4MN8), in grey. (**b**) LRR1 model showing secondary structure and solvent accessible surface area. (**c**) LRR1 model with AlphaFold predicted binding of alpha (in yellow) and gamma (in blue) flg22 epitopes displaying a putative binding pocket on LRR1 with solvent accessible area shown. (**d**) Hydrogen bonds between gamma residues (in blue) and LRR1 (in grey). Hydrogen bonds were predicted between Val-146, Val-148 and Arg-8; Val-148, Thr-150 and Asn-10; Thr-150 and Ala-12; and Arg-152 and Ser-11 with residues labelled. All hydrogen bonds were predicted using the PyMOL find polar contacts between chains function and are indicated by red dotted lines with lengths between 2.8 and 3.2 Å.

**Figure 6 plants-11-01475-f006:**
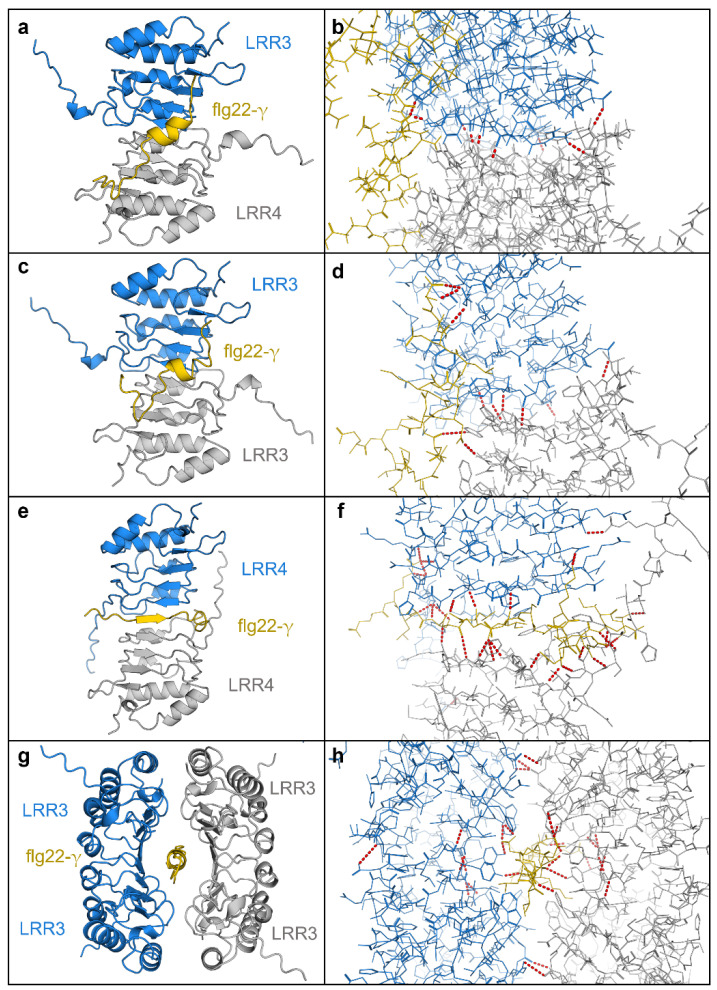
AlphaFold multimer complexes with *Brachypodium* LRRs. (**a**) The LRR from *BdIRI3* (LRR3) in blue, *BdIRI4* (LRR4) in grey, and flg22-y in yellow. (**b**) Hydrogen bonds in the LRR3 and LRR4 heterodimer and flg22-y complex. (**c**) Two monomers of LRR from *BdIRI3* (LRR3) in blue and grey, and flg22-y in yellow. (**d**) Hydrogen bonds in the LRR3 homodimer and flg22-y complex. (**e**) Two monomers of LRR from *BdIRI4* (LRR4) in blue and grey, and flg22-y in yellow. (**f**) Hydrogen bonds in the LRR4 homodimer and flg22-y complex. (**g**) Two homodimers of LRR3, in blue and grey, for a total of four LRR3 monomers in complex with flg22-y, in yellow. (**h**) Hydrogen bonds in the LRR3 homotetramer and flg22-y complex. All hydrogen bonds were predicted using the PyMOL find polar contacts between chains function and are indicated by red dotted lines with lengths between 2.8 and 3.2 Å.

## Data Availability

Not applicable.

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
