# Peer review of "Brachypodium Antifreeze Protein Gene Products Inhibit Ice Recrystallisation, Attenuate Ice Nucleation, and Reduce Immune Response"

_plants, 2022, doi:10.3390/plants11111475_

Round 1
Reviewer 1 Report
Cold stress has been reported to result in significant damage and yield loss in plants. In addition, chilling stress is becoming a great threat to crop production and quality. In this manuscript, the authors studied the function of BdAFPs in the mediation of chilling stress. The data showed that the AFP protein plays a key role in the decrease of INP-Induced Freezing damage. More interestingly, flg22 induced ROS accumulation was compromised in BdIR1s over-expression lines, as compared to WT.
To sum up, the manuscript was well organized and well written. Moreover, the authors present a very interesting topic. There are currently many interests in the improvement of cold tolerance in tiff brome. It would be of wide interest to the plant community, the crop industry, and the “Plants” readers. However, I have some concerns about the manuscript, before publication:
1. Some results were not fully explained. For example, authors did not explain why over-expression of IRI gene, which encoded LRRs protein, caused the suppression of plant immune responses.
2. As the abbreviation was used for the first time, the authors should explain it. Such as on page 2 line 62 “FLS”. Please check all of the abbreviations in the manuscript.
3. In the method section, page 13 line 435. Please check the description “OD600 ≥ 12”.
Author Response
Cold stress has been reported to result in significant damage and yield loss in plants. In addition, chilling stress is becoming a great threat to crop production and quality. In this manuscript, the authors studied the function of BdAFPs in the mediation of chilling stress. The data showed that the AFP protein plays a key role in the decrease of INP-Induced Freezing damage. More interestingly, flg22 induced ROS accumulation was compromised in BdIR1s over-expression lines, as compared to WT.
To sum up, the manuscript was well organized and well written. Moreover, the authors present a very interesting topic. There are currently many interests in the improvement of cold tolerance in tiff brome. It would be of wide interest to the plant community, the crop industry, and the “Plants” readers. However, I have some concerns about the manuscript, before publication:
1. Some results were not fully explained. For example, authors did not explain why over-expression of IRI gene, which encoded LRRs protein, caused the suppression of plant immune responses.
2. As the abbreviation was used for the first time, the authors should explain it. Such as on page 2 line 62 “FLS”. Please check all of the abbreviations in the manuscript.
3. In the method section, page 13 line 435. Please check the description “OD600 ≥ 12”.
- We thank the reviewer very much for this suggestion and in response we have revised the manuscript, including a revision in the Discussion to provide a more detailed explanation as follows:
“We speculate that the apoplast LRRs function as pattern receptor-like proteins, except that they lack the intracellular domain. With their affinity for flagellin, these LRRs are hypothesised to bind to flagella from P. syringae and other pathogens and therefore interfere with subsequent binding to FLS2. Such LRR occupation of the flagella would then reduce flagella availability for binding FLS2 and attenuate downstream effects involving BAK1, BIK1, and the phosphorylation of respiratory burst oxidase homolog protein D (RBOHD) that normally leads to extracellular release of ROS. ROS not only targets invading pathogens, but may also damage host tissues (Figure 4b,c). Therefore, LRR would mute the Brachypodium immune response and its attendant energetic requirements and thus joins other negative regulators of ROS production [43]. Indeed, compared to control plants, BdIRI knockdowns showed more cellular death post-infection and an overall higher susceptibility to pathogenic P. syringae [8].”
Furthermore, we have now added an illustration to Figure 4 to help readers visualize the hypothesized method of action, The figure caption has been modified accordingly.
- The first mention of FLS2 in the text has been defined as suggested
- This is correct according to the referenced paper (Zhang et al., 2020). This reference reads “…diluted 10 times to measure the OD600 (should be equal to or greater than 12).”
We have also made revisions to the reference list and added a new reference and with the addition to the figure, it should increase the clarity of the results.
Reviewer 2 Report
It is remarkable that the Brachypodium genes play multiple distinct roles in connecting freeze survival and anti-pathogenic systems. These roles are played through the ability of their encoded proteins to adsorb to ice, as well as through their ability to attenuate bacterial ice nucleation and the host immune response. Overall, this phenomenon is remarkable. Manuscript is well written and study design is OK. The manuscript can be published after a careful English language check.
Author Response
It is remarkable that the Brachypodium genes play multiple distinct roles in connecting freeze survival and anti-pathogenic systems. These roles are played through the ability of their encoded proteins to adsorb to ice, as well as through their ability to attenuate bacterial ice nucleation and the host immune response. Overall, this phenomenon is remarkable. Manuscript is well written and study design is OK. The manuscript can be published after a careful English language check.
Thank you to the reviewer for this suggestion. We have been careful to use all English spelling rather than US spelling for words with this option. This reviewer may well appreciate the addition of the new illustration to Figure 4. We agree with the reviewer’s view that these Brachypodium genes are remarkable!